# Dental Care Services for Older Adults in Hong Kong—A Shared Funding, Administration, and Provision Mode

**DOI:** 10.3390/healthcare9040390

**Published:** 2021-04-01

**Authors:** Stella Xinchen Yang, Katherine Chiu Man Leung, Chloe Meng Jiang, Edward Chin Man Lo

**Affiliations:** Faculty of Dentistry, The University of Hong Kong, Hong Kong, China; xcyang@connect.hku.hk (S.X.Y.); kcmleung@hku.hk (K.C.M.L.); cmjiang@hku.hk (C.M.J.)

**Keywords:** dental care, oral healthcare service, service mode, older adult, mode of finance

## Abstract

Hong Kong has a large and growing population of older adults but their oral health conditions and utilization of dental services are far from optimal. To reduce the financial barriers and to improve the accessibility of dental care services to the older adults, a number of programmes adopting an innovative shared funding, administration, and provision mode have recently been implemented. In this review, an online search on the Hong Kong government websites and the electronic medical literature databases was conducted using keywords such as “dental care,” “dental service,” and “Hong Kong.” Dental care services for older adults in Hong Kong were identified. These programmes include government-funded outreach dental care service provided by non-governmental organizations (NGOs), provision of dentures and related treatments by private and NGO dentists supported by the Community Care Fund, and government healthcare vouchers for private healthcare, including dental, services. This paper presents the details of the operation of these programmes and the initial findings. There is indirect evidence that these public-funded dental care service programmes have gained acceptance and support from the government, the service recipients, and the providers. The experience gained is of great value for the development of appropriate dental care services for the older adults in Hong Kong and worldwide.

## 1. Introduction

Located on the south coast of China, Hong Kong is one of the special administrative regions of the People’s Republic of China. It has a total population of 7.5 million and ranks fourth in the world in terms of population density (6782 residents per square kilometer) [1]. Hong Kong has the fourth highest human development index score and the life expectancy of Hong Kong people, 84.9 years, is the highest in the world [2]. With a low fertility rate of less than 2100 per 1000 women over the past decades [3], both the number and the proportion of older adults (people aged 60 or above) in Hong Kong have been increasing. At present, there are approximately 1.9 million people aged 60 or above residing in Hong Kong. According to the government projection, the proportion of older adults in the Hong Kong population will increase from 26.2% in 2020 to 37.7% in 2040 [4]. It is further projected that by 2050, the old-age dependency ratio will reach 71 persons aged 65+ years per 100 persons aged 20–64 years, making Hong Kong ranked among the top 10 countries or areas in the world [5].

Oral health is an integral part of general health. It affects chewing, nutrition intake, communication and participation in social activities. Oral health status of older adults can influence their quality of life and well-being [6]. Although Hong Kong is an economically highly developed city, the oral health status of its older adult population is far from satisfactory. The latest territory-wide oral health survey conducted by the government Department of Health in 2011 found a high prevalence of untreated active dental caries affecting 47.8% and 55.2% of the community-dwelling and the institutionalized older adults, respectively, while the respective mean numbers of decayed, missing, and filled teeth (DMFT) were 16.2 and 25.9 [7]. Regarding their periodontal health conditions, nearly all of the dentate community-dwelling older adults had bleeding gums and nearly half (48.1%) of them had periodontal pockets of 4 mm or more.

Oral diseases are common among the older adults in Hong Kong, indicating a high normative need for professional dental care services. However, their utilization of dental care services is suboptimal. In the 2011 Hong Kong Oral Health Survey, less than 40% of the surveyed older adults who suffered from severe toothache that disturbed sleep sought help from dentists [7]. Less than a quarter (22.3%) of the surveyed community-dwelling older adults and nearly none of the institutionalized older adults had a habit of regular dental check-up. Their most commonly reported access barrier was the financial burden related to dental treatments.

To tackle the oral health problems of the older adults in Hong Kong and to improve their access to proper dental care services, a number of dental care service programmes adopting an innovative shared funding, administration, and provision mode have recently been implemented. This article aimed to describe the scope of service, administration, and financial arrangements of the dental care service programmes, especially those that use public funding, so as to provide an overview of the dental care service for older adults in Hong Kong and to discuss the improvements needed.

## 2. Data Collection

The information on dental care services available for older adults in Hong Kong was mainly collected from publicly available information and publications. First, keywords “dental care” and “dental service” were used to search the Hong Kong Special Administrative Region (HKSAR) government websites. Second, keywords “oral healthcare in Hong Kong,” “dental care in Hong Kong,” and “dental service in Hong Kong” were used for a general search of the online database and the electronic medical literature databases. Dental care services available for older adults in Hong Kong were identified. Supplementary data of the services were collected by searching the websites of the specific dental programmes, reviewing publicly available reports and consulting experts.

## 3. Private Dental Care Service for the Older Adults in Hong Kong

There are approximately 2500 registered dentists in Hong Kong in 2020, giving a dentist to population ratio of 1:3000 (33 dentists per 100,000 population). Since Hong Kong adopts a predominantly private free-market economic model, dentists are free to choose where they practice and to decide on their treatment charges. More than 80% of the registered dentists are working in the private sector, mostly in solo or small group practices. Their treatment charges vary greatly and dental fees are generally regarded as high by the public [8]. Private dentists are the main dental service providers for the older adults. In the 2011 Hong Kong Oral Health Survey, two-thirds (68%) of the dental visits of the surveyed community-dwelling older adults were made to private dentists [7]. Dental third party payment or insurance coverage for retired persons and older adults is uncommon in Hong Kong. Although the free-market model adopted in the private sector provides a wide range of choices of providers and type of dental care for the consumers, this only works well for the people who can afford to pay for the high treatment fees.

Economically, Hong Kong is a high income area, e.g., the gross national income per capita in 2019 was USD 63,000 [2]. Nonetheless, around one-third of older adults are living in relative poverty [9], and they are a financially vulnerable group. A recent government thematic household survey revealed that dental care services utilization in Hong Kong was positively associated with monthly household income, and only 37% of the adults aged 65 or above had visited a dentist in the previous year [10]. Since most of the older adults in Hong Kong are not economically active and around one-third of them live below the poverty line [8], it is important to develop strategies to provide adequate, appropriate, and accessible dental care services for those who cannot afford private dental care service.

## 4. Public-Funded Dental Care Service for the Older Adults in Hong Kong

Hong Kong government adopts a shared funding, administration, and provision model for its healthcare services, including dental care services for the older adults [11]. The funding sources include the government general revenue and charity foundations while the administration and providers include the government, non-profit-making non-governmental organizations (NGOs), and private dentists (Figure 1). Details of the public-funded dental care service programmes are described below.

### 4.1. Dental Service Funded and Provided by the Government

#### Emergency and Hospital in-Patient Dental Services

There are approximately 40 government dental clinics in Hong Kong and they mainly provide comprehensive dental care service to civil servants and their dependents, and pensioners. Only 11 government dental clinics provide limited emergency dental services to meet the demand of the general public (Table 1) [12]. However, the services are restricted to management of acute dental problem, e.g., teeth extraction (one tooth per visit), medication to relieve dental pain and treatment of oral abscess. Moreover, these 11 clinics are unevenly distributed in the 18 districts of Hong Kong. Furthermore, the allocated time for providing free emergency dental treatment to the public is rather limited, two of these 11 clinics allocate one morning per month and the other clinics allocate one or two morning/afternoon sessions per week. The maximum number/quota of patients to be treated in a session in a clinic ranges from 32 to 84. In 2017–2018, 35,957 persons used this service, among whom 57% were aged 61 or above [13].

In addition, older adults in Hong Kong are eligible to receive oral and maxillofacial surgery as an in-patient in seven public hospitals through referral by private or public dentist/doctors [14]. Provision of treatment in public hospitals is arranged according to the urgency of patient’s condition. The cost of the dental specialist treatments is heavily (>90%) subsidized by the government and the patient only needs to pay a nominal fee.

### 4.2. Dental Service Funded by the Government and Provided by NGOs

#### 4.2.1. Outreach Dental Care Programme (ODCP) for the Older Adults

Aiming to promote prevention and early treatment of oral diseases, and to enhance the oral hygiene of the older adults using long-term care (LTC) social welfare services in residential care homes or day care centers, the ODCP was launched in 2011 by the Hong Kong government as a 3-year pilot scheme and was later converted into a recurrent programme in 2014 (Table 1) [15]. The ODCP is fully funded by the Department of Health of the Hong Kong government but it is administered mainly by NGOs which employ dental health workers to provide the clinical service (Figure 2). The scope of the dental care service has been expanded from just on-site dental check-up, scaling, pain relief, and emergency treatments in the pilot scheme to the additional provision of other curative treatments such as filling, tooth extraction, and dentures in the present programme. Older adults in need of dental treatments that are difficult to be provided onsite in the residential care home or day care center, such as X-ray examination, can also receive the treatments in the dental clinics of the NGOs. On-site oral care training and oral health education activities such as seminars/talks are provided to the older adults, their family members and caregivers.

NGOs that operate dental clinics as part of their services can apply for government financial support to set up outreach dental service teams. An outreach dental team usually consists of one dentist, one dental surgery assistant, and a general helper. The participant NGO and its outreach dental team are responsible to liaise with the LTC facilities under its management (if any) and those in the geographical districts assigned by the government. The assignment of LTC facilities to different NGO outreach dental teams helps to ensure all older adults using LTC facilities in Hong Kong are covered by the programme and that the participant NGOs do not need to compete to serve LTC facilities.

The NGOs need to liaise with the LTC facilities regarding the logistic arrangements, such as transportation of dental equipment, and the date and time of the outreach dental visits. Usually a multi-purpose room in the LTC facility is set up as a temporary site for the provision of dental service on the day of the visit. When further treatments in a fully equipped dental clinic are needed, subsidized transportation and escort services can be arranged to bring the ODCP recipients to the dental clinics run by the same NGO that provides the outreach service. The need for these clinic-based dental treatments, such as fillings, tooth extractions, and denture work, will be assessed by assigned government dentists and if approved, the treatment cost will be covered by the government as part of the ODCP.

In the pilot scheme in 2011 to 2013, the cumulative project cost was HK$ 63 million (USD 0.8 million, 1 USD = HK$ 7.8) and there were 24 NGO outreach dental teams serving a total of 66,000 older adults [16]. At the close of the pilot scheme, around 70% of the approached older adults participated in the ODCP and received onsite dental care service. A total of 863 oral health education talks and training sessions were delivered to the older adults, their family members and caregivers.

In the recurrent ODCP, 23 outreach dental teams were set up by 10 NGOs. In the period from October 2014 to January 2020, about 233,700 older adults in LTC facilities had received the outreach dental care service [17]. The current funding mode is a combination of per-capita and per-item arrangement in which the government pays a fixed amount (approximately HK$ 500 or USD 65) per older adult served on-site disregarding the amount of service delivered. In addition, for the approved dental treatments delivered, the government pays the NGO according to the quantity (number of items) and type of dental treatment (different fixed fee for different defined treatment) provided. Under this mode of funding, the dental service providers have incentives both to serve a larger number of older adults and to provide the more complex dental treatments. On the other hand, the government can maintain control over the programme expenditure by determining the per-capita subsidy and the fee for each additional dental treatment item. Additionally, the provision of dental treatments in a dental clinic, which involves additional financial subsidy, requires prior approval by a government dentist who assesses the necessity of the treatment to avoid abuse of the programme.

#### 4.2.2. Comprehensive Social Security Assistance Dental Grant

The Comprehensive Social Security Assistance (CSSA) scheme was set up by the Social Welfare Department of the Hong Kong government with the purpose to bring the income of the individuals and families in financial hardship up to a prescribed level to meet their basic needs [18]. The CSSA scheme is non-contributory but means-tested, and the applicant must go through both income and asset assessments. CSSA recipients aged 60 years or above are eligible to apply for a special dental grant to cover the cost of necessary dental treatments. The treatment items covered by this grant include tooth extraction, scaling, filling, root canal treatment, denture, and fixed dental prosthesis (crown and bridge).

To apply for the dental grant, an older adult needs to visit one of the designated NGO dental clinics for a clinical examination and obtain a dental treatment plan with an estimation of the treatment cost. After approval of the application by the Social Welfare Department, the older adult can seek dental service from any registered dentist, pay for the treatment and then submit the relevant receipts for reimbursement. The amount reimbursed is determined by the actual expense on the dental treatments received, the cost estimation provided by the designated NGO dental clinic, and the ceiling amounts of the various dental treatment items set by the government under this scheme (Table 1). Although the CSSA recipients can seek treatment from private dentists, they usually visit the designated NGO dental clinics to receive the approved treatments so as to avoid the need to top up the charges out of their own pocket.

### 4.3. Dental Care Service Funded by the Government and Provided by NGOs and Private Dentists

#### 4.3.1. Community Care Fund Elderly Dental Assistance Programme

The Community Care Fund (CCF) was established in 2011 and is administered by the Commission on Poverty chaired by the government Chief Secretary for Administration [19]. With regular monetary injections mainly from the government, the CCF provides assistance to Hong Kong citizens who have financial difficulties but are not covered by the social welfare safety net. Financed by the CCF, the Elderly Dental Assistance Programme was launched in 2012 and is designed to serve low-income older adults who suffer from eating or chewing problems due to dental cause. The Hong Kong Dental Association (the dental professional body of Hong Kong) administers this programme [20]. Operational guidelines are drawn up, and dentists working in private practices and NGO dental clinics can apply to become a recognized service provider under this programme.

The initial target population of the programme was older adults aged 60 years or above who were non-CSSA recipients but were covered by the subsidized community or home care services schemes subvented by the Social Welfare Department. The target population of the programme was later expanded to also cover older adults who were recipients of the Old Age Living Allowance, a means-tested scheme for older adults aged 65 years or older in need of financial support [21].

To participate in this programme, the eligible older adults can directly contact the social service agencies from which they are receiving care or the dentists registered with this programme [22]. Each beneficiary older adult can receive a maximum subsidy of HK$ 15,350 (USD 1970) for the dental service received (Table 1). The subsidized dental service covers removable dentures and the denture-related treatments, including oral examination and radiographs, scaling, tooth extraction, and filling [23]. By the end of January 2020, a total of 71,440 older adults had participated in this programme and 82% of them had their required dental treatments completed by 564 private dentists and in 69 NGO dental clinics [17].

#### 4.3.2. Elderly Health Care Voucher Scheme

The Elderly Health Care Voucher Scheme (HCVS) administered by the government Department of Health aims to encourage utilization of private primary healthcare services, including dental service, by the older adults [24]. Registered dentists, as well as healthcare professionals from nine other disciplines, can join the HCVS by enrolling with the government. A logo of the scheme is put up outside the clinic of the enrolled healthcare provider for easy identification. All older adults aged 65 years or above with a valid Hong Kong identity card are eligible to join the scheme without the need for pre-registration. After having received dental treatments from a dentist working in a private or NGO dental clinic, the older adult patient only needs to show his/her identity card, sign a consent form for the dentist to use his/her personal data for administration purpose, and sign on a form indicating the voucher amount to be deducted. The amount of e-voucher was HK$ 250 (USD 32) a year when it was first launched in 2009 and has increased by phases to the present amount of HK$ 2000 (USD 256) a year (Table 1). The unspent voucher can be carried forward to the following years with a ceiling of HK$ 8000 (USD 1026). There are no limits on the expenditures in each healthcare visit and no expiry date of the unspent voucher. The older adults have a free choice of private healthcare services and providers. The total voucher amount claimed by dental service providers increased from HK$ 105 millions (USD 14 millions) in 2016 to HK$ 313 millions (USD 40 millions) in 2019 [25].

## 5. Discussion

Judging from the developments of the above-described dental care service programmes for older adults in Hong Kong, it seems that all of the programmes are generally accepted by and has received continue support from the government (source of funding), the NGOs and private dentists (the administrators and dental service providers) and the older adults (the consumers). These developments include the continue increase in public funding, the expansion of service scope of the ODCP and the CCF Elderly Dental Assistance Programme, and the increase in the amount of the HCVS spent on dental care services. Despite these, information on whether there are improvements in the oral health of the Hong Kong older adults, especially those who have used these programmes, is limited and needs to be collected in future epidemiological studies. Territory-wide surveillance oral health surveys had been conducted in Hong Kong once every 10 years since 1991 [7] and the next survey is planned to be conducted in 2021.

Unlike many of the countries in northern and western Europe where there is public-funded dental care service covering all older adults [26], the dental service programmes in Hong Kong are only offered to selected groups of older adults with special needs. Some aspects of the funding and operation modes of these Hong Kong dental programmes are similar to those of the programmes in other economically developed countries. For example, the CCF Dental Assistance Programme in Hong Kong sets a fixed fee for each dental treatment item as a way of financial and administrative control which is similar to the practice of the National Health Service (NHS) in the UK. In Hong Kong, the CSSA provides a social welfare safety net for the low income older adults and the CSSA dental grant helps them to access necessary dental care services, which is comparable to the Medicare Advantage Plans in the USA [27].

Since Hong Kong adopts a capitalistic and a free-market economic system, financially vulnerable older adults who cannot afford to pay for private dental treatments can benefit from the recently implemented public-funded dental care service programmes. On the one hand, there are some overlaps of the target population of these programmes. For example, the recipients of the CSSA dental grant can also use their e-vouchers from the HCVS to pay for their dental treatments received in private dental clinics. On the other hand, the coverage of some of these programmes is mutually exclusive. For example, the older adults who have participated in the ODCP cannot apply for the CCF Elderly Dental Assistance Programme. Since the ODCP is an annual onsite dental service while the CCF is provided on an “once in a lifetime” basis, older adults may be confused and have difficulty in choosing the appropriate programme. It should be noted that for older adults in Hong Kong who have non-financial obstacles in seeking dental care, such as difficulty in accessing a dental clinic due to physical constraints, only the ODCP may be helpful as it provides on-site dental care service and offers additional subsidy for escort service to the NGO dental clinics. More coordination between the public-funded dental care service programmes in Hong Kong are needed. Besides, more information and guidance should be provided to the older adults in Hong Kong so that they can make appropriate decisions, based on their individual conditions, on which dental care programme(s) they should use.

As for the service scope, most of the dental care service programmes for the older adults in Hong Kong focus on provision of emergency and curative dental treatments. Only the ODCP provides regular preventive dental care such as annual dental examinations and topical fluoride applications, which is a basic component of the dental service in some overseas healthcare programmes such as the NHS in the UK and the Medicare in the USA [26]. The Hong Kong dental programmes should be enhanced by including more oral health education and prevention-oriented services for better long-term oral health promotion.

The poor oral health status and low utilization of professional dental service among the older adults has attracted the attention of the Hong Kong government and a shared funding, administration, and provision mode is used in the recently introduced dental care service programmes. A reason for this is the relatively small number of dentists in Hong Kong as compared to other countries and places with similar economic status. As the Hong Kong government only employs approximately 15% of the dentists in Hong Kong, it has to make use of the dental workforce in the NGOs and private sector to provide dental service to the older adults, especially those with financial and access difficulties, in the public-funded dental care service programmes. With continue expansion of the service scope and the eligible population, and increase in the amount of subsidies in these programmes, the number of older adults in Hong Kong benefitting from this shared mode of dental service provision has been increasing continuously [17].

Since financial difficulty is a common barrier preventing older adults from accessing necessary dental care [7,28], the dental subsidies and free dental treatments provided in the recently introduced public-funded dental care service programmes in Hong Kong can be expected to encourage more older adults to utilize dental care services for improvement of their oral health. Like in many places, shortage of dental manpower and lack of expertise in geriatric dentistry are limiting factors for the development of dental services for older adults in Hong Kong [9,29]. To address these issues, the Hong Kong government has recently supported an increase in the annual intake of undergraduate dental students from 53 to 80 into the only local dental school. However, it will take a number of years to see a substantial increase in the supply of locally trained dentists because the dental degree programme run by the University of Hong Kong is a 6-year curriculum. Other than increasing the intake of first-degree dental students and attracting overseas dentists to practice in Hong Kong, training in geriatric dentistry should be provided to the less experienced dentists to fulfill the surge of demand for dental care service by the older adults in the coming years. Postgraduate programmes should be designed to meet this need. It is also important to recognize geriatric dentistry as a dental specialty and to set up dental clinics which are specially equipped for treating frail or wheelchair-bound older adults [30].

To improve the dental care services for older adults, enhancement of the collaboration between dental and other primary healthcare personnel, and provision of oral health care training for other healthcare professionals have been suggested [6]. However, due to the historical development of the medical and dental professions, and of the healthcare services in Hong Kong, dental services are separated from the medical services in the general healthcare system. An integrated approach in the further development of primary health care services for the ageing population has yet to be developed. More work on the incorporation of dental health into the promotion of general health of the older adults is needed.

## 6. Conclusions

A number of public-funded dental care service programmes for older adults has recently been implemented in Hong Kong using a shared funding, administration, and provision mode. There are some indirect evidences showing that these programmes are generally accepted and supported by the government, the NGOs and private dentists, and the older adults. The experience gained is of great value for the future development of appropriate dental care service for the older adult population in Hong Kong and worldwide.

## Figures and Tables

**Figure 1 healthcare-09-00390-f001:**
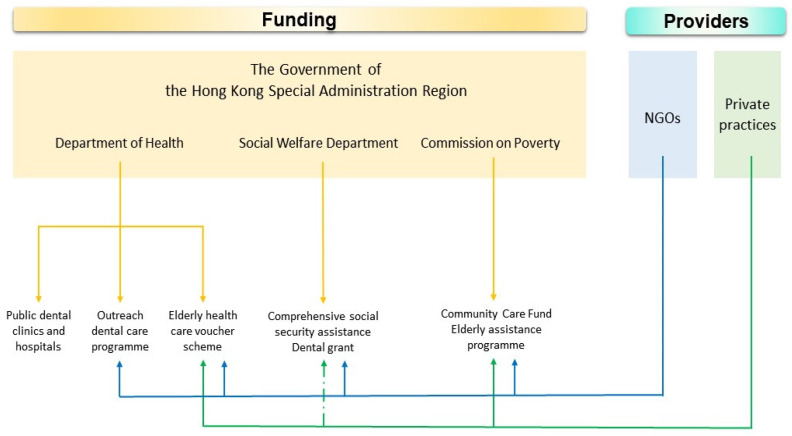
Overview of the public-funded dental care service programmes for older adults in Hong Kong.

**Figure 2 healthcare-09-00390-f002:**
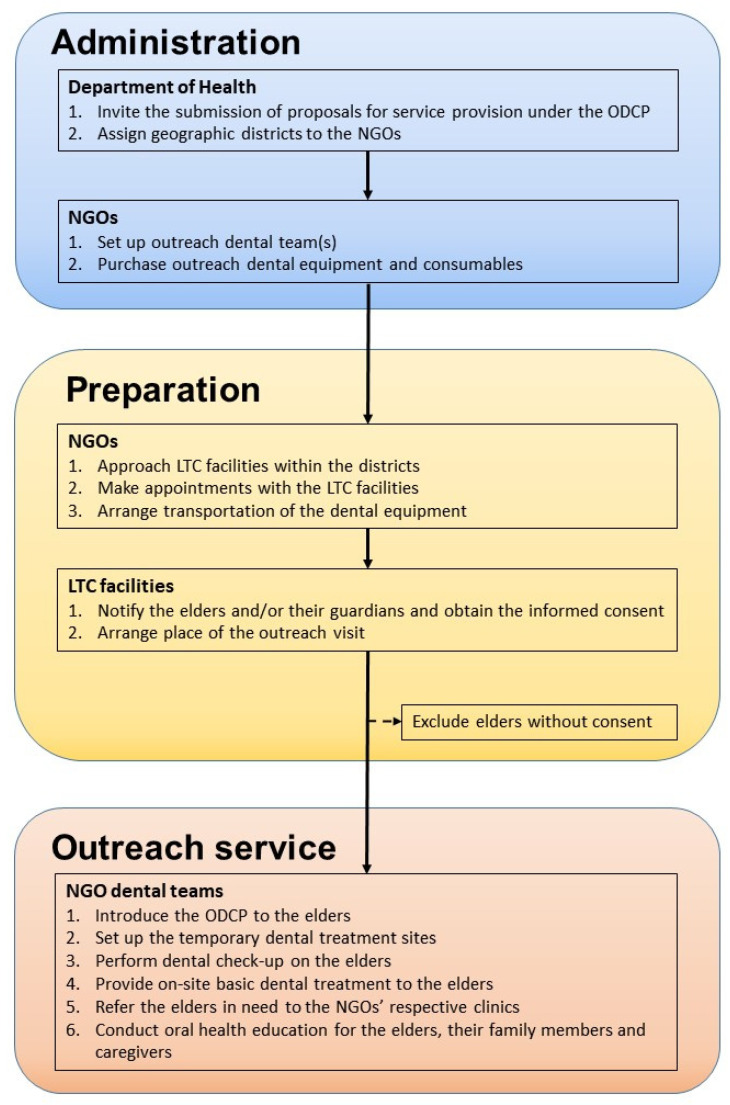
The flow of the activities of the outreach dental care programme for older adults.

**Table 1 healthcare-09-00390-t001:** Summary of public-funded dental care services for the older adults in Hong Kong.

Aspects	Government Dental Service	Outreach Dental Care Programme (ODCP)	Comprehensive Social Security Assistance (CSSA) Dental Grant	Community Care Fund (CCF) Elderly Dental Assistance Programme	Elderly Health Care Voucher Scheme (HCVS)
**Establishment**	Since 1950	Since 2011	Since 1980	Since 2012	Since 2009
**Eligibility**	Persons with emergency dental treatment needs Public hospital in-patients with dental treatment needs	Older adults residing in residential care homes or using day-care service	CSSA recipients aged 60 years or above	Older adults aged 60 years or above, non-CSSA recipients but are covered by the subsidized community or home care services schemes Recipients of the Old Age Living Allowance	Older adults aged 65 years or above
**Required to pass a means-test**	No	Yes	Yes	Yes	No
**Service scope**	Emergency dental care, such as tooth extraction, pain relief and treatment of oral abscess	Oral health education On-site basic dental care, including dental check-up, scaling and emergency treatments Further treatments in dental clinic, including filling, tooth extraction, and denture	Tooth extraction, scaling, filling, root canal treatment, denture, and fixed dental prosthesis	Removable denture and denture-related services	Private primary healthcare services, including dental service
**Service providers and venue**	Government dentists; 11 government dental clinics and 7 public hospitals	Private dentists working for NGOs; LTC facilities and NGO dental clinics	Private dentists; Designated NGO dental clinics and private dental clinics	Private dentists; NGO and private dental clinics	Private dentists; NGO and private dental clinics
**No. of beneficiary in a year ***	20,164	44,514	Not reported	9746	Not reported
**Amount of subsidy per user ****	Not applicable	Onsite service: HK$ 500 Further treatment: maximum amount set for each treatment item	A maximum amount set for each approved treatment item	Maximum: HK$ 15,350	HK$ 2000 per year; Can accumulate up to a ceiling of HK$ 8000

* Estimates based on the reports of the programmes. ** Exchange rate: 1 USD = HK$ 7.8.

## Data Availability

Not applicable.

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
