# Peer review of "Dental Care Services for Older Adults in Hong Kong—A Shared Funding, Administration, and Provision Mode"

_healthcare, 2021, doi:10.3390/healthcare9040390_

Round 1

Reviewer 1 Report

I congratulate the authors for this interesting and timely article regarding dental care services for elders in Hong Kong. The organization of the paper is appropriate. This paper is very well written and has interesting figures.

This is a review paper on the dental care services for older adults in Hong Kong. The paper describes in detail the various provisions and services implemented  for the older population over the past decade. 

However, there are numerous language corrections to be made throughout the manuscript. 

Author Response

Comment: I congratulate the authors for this interesting and timely article regarding dental care services for elders in Hong Kong. The organization of the paper is appropriate. This paper is very well written and has interesting figures. This is a review paper on the dental care services for older adults in Hong Kong. The paper describes in detail the various provisions and services implemented for the older population over the past decade.

Response: We thank the reviewer for the positive comments.

Comment: However, there are numerous language corrections to be made throughout the manuscript.

Response: The manuscript has been thoroughly checked and edited for language.

Reviewer 2 Report

Dear authors,

The topic of the paper is really original and little addressed among the elderly population. Although it seems important to me to improve the quality of life and health of the elderly, I must indicate that the manuscript has severe problems.

Below you can see my comments:

  1. Abstract and keywords: The abstract includes the objective of the work but not the methodology carried out to achieve it. The results are imprecise: “There is evidence that these public funded dental care service programs have gained acceptance and support from both the service recipients and the providers”. Could you indicate what that evidence is?. Regarding keywords, despite highlighting the financing of the programs in the manuscript, no keyword on this aspect is included.
  2. Objective: "To describe Hong Kong dental care service programs, especially those that use public funding, so that the readers can have more knowledge and ideas on the provision of dental healthcare to the elders in different places". The objective is very general, so it would have been interesting to present specific objectives or at least indicate what aspects or variables that description of the programs includes. Do you mean the description of the quality perceived by the users, the demonstrated efficacy of these programs, barriers or conflicts in their implementation?
  3. Introduction / Background: The introduction is adequate, and justifies the work. Although you use data from an outdated survey "The oral health survey conducted by the government Department of Health in 2011", you also mention "A government thematic household survey conducted in 2019". Regarding section 2. "Private dental care service for the elders in Hong Kong", I think it duplicates information already mentioned in the introduction. I would delete this section and only add new information to the first section described by the authors.
  4. Methodology: Where is the methodology? The absence of methodology is a very serious problem. I believe that it is not possible to publish any article if the authors are not rigorous. Are the authors sure that all dental care programs or models are included in this work? How do you know all these programs? How have you analyzed their characteristics? For example: “There are around 40 government dental clinics in Hong Kong”. Have they used a mapping system to identify dental clinics? Have you conducted interviews with key agents to learn more about how these programs work? Where do they get the information to describe these programs?
  5. Discussion: The discussion is unrelated to the results, since instead of focusing on dental care programs and their financing, it seems to focus on strategies to increase the number of providers (dentists), such as increasing the number of students and developing postgraduate studies with a geriatric dentistry perspective. I would have liked to read a comparison of programs with other similar population groups and other geographical areas.
  6. Conclusions: “There are some initial evidences showing their success in terms of participant and provider acceptance”. I do not find this evidence in the results described by the authors. They are based on a very basic description of these programs.

Author Response

Comment: Abstract and keywords: The abstract includes the objective of the work but not the methodology carried out to achieve it.

Response: In the revised abstract, description of the methodology used to obtain information on the dental care service programmes for older adults in Hong Kong is added (see page 1).

Comment: The results are imprecise: “There is evidence that these public funded dental care service programs have gained acceptance and support from both the service recipients and the providers”. Could you indicate what that evidence is. Regarding keywords, despite highlighting the financing of the programs in the manuscript, no keyword on this aspect is included.

Response: The indirect evidence used for supporting the conclusion statements are presented and discussed (see page 7, paragraph 2). “Mode of finance” is added to the list of keywords.

Comment: Objective: "To describe Hong Kong dental care service programs, especially those that use public funding, so that the readers can have more knowledge and ideas on the provision of dental healthcare to the elders in different places". The objective is very general, so it would have been interesting to present specific objectives or at least indicate what aspects or variables that description of the programs includes. Do you mean the description of the quality perceived by the users, the demonstrated efficacy of these programs, barriers or conflicts in their implementation?

Response: This article aimed to describe the scope of service, administration and financial arrangements of the dental care service programmes, especially those that use public funding, so as to provide an overview of the dental care service for older adults in Hong Kong and to discuss the improvements needed. The revised objective is presented in page 2, paragraph 3.

Comment: Introduction / Background: The introduction is adequate, and justifies the work. Although you use data from an outdated survey "The oral health survey conducted by the government Department of Health in 2011", you also mention "A government thematic household survey conducted in 2019". Regarding section 2. "Private dental care service for the elders in Hong Kong", I think it duplicates information already mentioned in the introduction. I would delete this section and only add new information to the first section described by the authors.

Response: We thank the reviewer for the suggestion. The duplicated information in the Introduction section is deleted and that in the Results section (Section 3) is kept (see page 2, paragraph 5).

Comment: Methodology: Where is the methodology? The absence of methodology is a very serious problem. I believe that it is not possible to publish any article if the authors are not rigorous. Are the authors sure that all dental care programs or models are included in this work? How do you know all these programs?

Response: This paper is a narrative review on the current dental care services for older adults in Hong Kong. Thus, the methods used in the collection of information was less stringent than those usually adopted in systematic reviews. A new section (Section 2) on the methods used in the online information search is added in the revised manuscript (see page2, paragraph 4).

Comment: How have you analyzed their characteristics? For example: “There are around 40 government dental clinics in Hong Kong”. Have they used a mapping system to identify dental clinics? Have you conducted interviews with key agents to learn more about how these programs work? Where do they get the information to describe these programs?

Response: The list of government dental clinics in Hong Kong is available on a government website (https://www.dh.gov.hk/tc_chi/clinictimetable/dc.htm) which is cited in the paper. Since one of the authors of this manuscript is an expert in dental public health and is a consultant of these public-funded programmes, we know the details of the implementation of these programmes. Information on these programmes is also available in publicly available reports or websites. We summarize the information and provide an overview in the Results. A new Table 1 is added.

Comment: Discussion: The discussion is unrelated to the results, since instead of focusing on dental care programs and their financing, it seems to focus on strategies to increase the number of providers (dentists), such as increasing the number of students and developing postgraduate studies with a geriatric dentistry perspective. I would have liked to read a comparison of programs with other similar population groups and other geographical areas.

Response: The discussions in this manuscript include aspects on the necessary improvements and the sustainability of the programmes. In the revised manuscript, we have added discussions on the comparison of the Hong Kong programmes with similar programmes in other economically developed countries, such as the target population, coverage and service scope of the programmes (see page 7, paragraphs 3 and 4).

Comment: “There are some initial evidences showing their success in terms of participant and provider acceptance”. I do not find this evidence in the results described by the authors. They are based on a very basic description of these programs.

Response: The indirect evidence used for supporting the conclusion statements are presented and discussed (see page 7, paragraph 2).             

Reviewer 3 Report

This is a review paper on the dental care services for older adults in Hong Kong. The paper describes in detail the various provisions and services implemented  for the older population over the past decade. 

There are grammatical and language errors throughout the paper, including in figure 2. For example, in the introduction (line 6), the use of the phrase 'Hong Kong people" could be changed to "people in Hong Kong". In Figure 2, under 'Preparation/LTC facilities: Notice the elders' may be corrected to 'Notify the older adults'. As there are numerous minor language errors throughout the paper, it will benefit from a detailed language editing and correction process.

The term "elderly" may be considered as stigmatizing/ ageist. The use of the term "elder", "elderly", can be replaced by the term "older adult" throughout the paper as well as in the title. "Elderly homes" may be replaced by "aged care homes", "retirement homes" etc. 

Certain sentences may have to be rephrased in order to provide more clarity on the idea that is being conveyed. For example:

"However, due to historical development of the medical and dental professions and the healthcare services in Hong Kong, dental services has been separated from the general healthcare services and an integrated approach has yet to be developed."- This sentence from paragraph 3 of the Discussion seems incomplete in context to the subject that is explained. 

"Elders in Hong Kong should have benefited from the recently implemented public funded dental care service programmes."- This sentence from paragraph 4 of the Discussion is an assumption. It can be rephrased appropriately. 

Author Response

Comment: This is a review paper on the dental care services for older adults in Hong Kong. The paper describes in detail the various provisions and services implemented for the older population over the past decade. There are grammatical and language errors throughout the paper, including in figure 2. For example, in the introduction (line 6), the use of the phrase 'Hong Kong people" could be changed to "people in Hong Kong". In Figure 2, under 'Preparation/LTC facilities: Notice the elders' may be corrected to 'Notify the older adults'. As there are numerous minor language errors throughout the paper, it will benefit from a detailed language editing and correction process.

Response: The manuscript has been thoroughly checked and edited for language. The language errors pointed out by the reviewer are corrected.

Comment: The term "elderly" may be considered as stigmatizing/ ageist. The use of the term "elder", "elderly", can be replaced by the term "older adult" throughout the paper as well as in the title. "Elderly homes" may be replaced by "aged care homes", "retirement homes" etc.

Certain sentences may have to be rephrased in order to provide more clarity on the idea that is being conveyed. For example:

"However, due to historical development of the medical and dental professions and the healthcare services in Hong Kong, dental services has been separated from the general healthcare services and an integrated approach has yet to be developed."- This sentence from paragraph 3 of the Discussion seems incomplete in context to the subject that is explained.

"Elders in Hong Kong should have benefited from the recently implemented public funded dental care service programmes."- This sentence from paragraph 4 of the Discussion is an assumption. It can be rephrased appropriately.

Response: The language errors are corrected according to the reviewer’s suggestions. The term “elderly” is replaced by “older adults” throughout the revised manuscript.

Reviewer 4 Report

Dear author,

i have carefully read yours article and to be honest I do not see any point why it was written.

First misleading point is the recongnizing this article as review. You are not doing any review of research or publishe articles. Most of yours refferences is pointing some websites notes and reports.

You are estimating the oral status of Hong Kong elders based on the report from 2011. Eventhough you are describing some programmes that the Hong Kong goverment founded, we do not see any results if there is improvment or not.

With the regretes I have to recommend the rejection of your article. Completely other situation will be if you will do methodicaly the same study as in 2011 and compare the rsults. Than, you will get amazing article with high impact. 

Author Response

Comment: I have carefully read yours article and to be honest I do not see any point why it was written. First misleading point is the recongnizing this article as review. You are not doing any review of research or published articles. Most of yours references is pointing some websites notes and reports.

Response: This paper is a narrative review on the current dental care services for older adults in Hong Kong. Thus, the methods used in the collection of information was less stringent than those usually adopted in systematic reviews. A new section (Section 2) on the methods used in the online information search is added in the revised manuscript (see page2, paragraph 4). The search results show that most of the information is only available in the Hong Kong government websites and reports instead in published scientific articles.

We think this manuscript is an important source of information for the healthcare researchers or policymakers in other countries and places to see how a shared funding, administration and provision mode is used in the provision of dental care services in Hong Kong. This manuscript can be a useful reference when the readers plan healthcare services for the older population in their home country.

Comment: You are estimating the oral status of Hong Kong elders based on the report from 2011. Even though you are describing some programmes that the Hong Kong goverment founded, we do not see any results if there is improvement or not. With the regrets I have to recommend the rejection of your article. Completely other situation will be if you will do methodicaly the same study as in 2011 and compare the results. Then, you will get amazing article with high impact.

Response: We agree with the reviewer that information on whether there are improvements in the oral health of the Hong Kong older adults, especially those who have used these programmes, is limited and needs to be collected in future epidemiological studies. Territory-wide surveillance oral health surveys had been conducted in Hong Kong once every 10 years since 1991 and the next survey is planned to be conducted in 2021. Discussion on this point is added in the revised manuscript (see page 7, paragraph 2).

Reviewer 5 Report

This is a narrative and qualitative piece of work discussing the dental health care provision in elderly in Hong Kong. No data are analysed but the current policy is discussed and how dental health care could be improved.

My understanding of HK is that the difference between rich and poor is very large (super billionaires versus very poor people). The authors may want to better discuss for whom the programs are best suited, what can be expected, what the average yearly dental costs are, what the average income is, and what improvement in oral health per dollar spent can be expected.

Unlike other social countries in Europe, HK is very expensive and capitalistic, favoring the rich and disadvantaging the poor. The authors may also want to elaborate on that. In northern European countries, for example, such a huge gap in oral health between rich and poor does not exist due to the social health care system. 

Author Response

Comment: This is a narrative and qualitative piece of work discussing the dental health care provision in elderly in Hong Kong. No data are analysed but the current policy is discussed and how dental health care could be improved.

My understanding of HK is that the difference between rich and poor is very large (super billionaires versus very poor people). The authors may want to better discuss for whom the programs are best suited, what can be expected, what the average yearly dental costs are, what the average income is, and what improvement in oral health per dollar spent can be expected.

Unlike other social countries in Europe, HK is very expensive and capitalistic, favoring the rich and disadvantaging the poor. The authors may also want to elaborate on that. In northern European countries, for example, such a huge gap in oral health between rich and poor does not exist due to the social health care system.

Response: We thank the reviewer for the suggestions on the enhancement of this manuscript. We agree that there is a huge gap between the dental care available for the rich and the poor in Hong Kong. Thus, the target population of the public-funded dental care programs are the unprivileged older adults in Hong Kong. In the revised manuscript, we have added discussions on the comparison of the Hong Kong programmes with similar programmes in other economically developed countries, such as the target population, coverage and service scope of the programmes (see page 7, paragraphs 3 and 4).